# Extent of compliance with COVID-19 prevention and control guidelines among supermarkets in Kampala Capital City and Mukono Municipality, Uganda

Richard K. Mugambe[1]*, Tonny Ssekamatte[1], Stevens Kisaka[2], Solomon T. Wafula[1], John Bosco Isunju[1], Aisha Nalugya[1], Patience Oputan[1], Douglas Kizito Makanga[3], Michael Mukiibi[4], Esther Buregyeya[1], Simon Kasasa[3], Winnifred K. Kansiime[1], Julie Balen[5], Renuka Kapoor[6], Joanne A. McGriff[6]

1 Makerere University School of Public Health, Kampala, College of Health Sciences, Makerere University, Kampala, Uganda, 2 College of Veterinary Medicine, Animal Resources and Biosecurity, Makerere University, Kampala, Uganda, 3 Ministry of Health, Kampala, Uganda, 4 World Health Organisation, Kampala, Uganda, 5 School of Health and Related Research, The University of Sheffield, Sheffield, United Kingdom, 6 The Center for Global Safe Water, Sanitation and Hygiene. Rollins School of Public Health, Emory University, Atlanta, GA, United States of America

* rmugambe@musph.ac.ug

**Data Availability Statement:** All relevant data are within the manuscript and its Supporting Information files.

## Abstract

### Background

Despite the development and enforcement of preventive guidelines by governments, COVID-19 continues to spread across nations, causing unprecedented economic losses and mortality. Public places remain hotspots for COVID-19 transmission due to large numbers of people present; however preventive measures are poorly enforced. Supermarkets are among the high-risk establishments due to the high interactions involved, which makes compliance with the COVID-19 preventive guidelines of paramount importance. However, until now, there has been limited evidence on compliance with the set COVID-19 prevention guidelines. Therefore, this study aimed to measure compliance with the COVID-19 prevention guidelines among supermarkets in Kampala Capital City and Mukono Municipality Uganda.

### Methods

A cross-sectional study was conducted among selected supermarkets in Kampala Capital City and Mukono Municipality in September 2020. A total of 229 supermarkets (195 in Kampala City and 34 in Mukono Municipality) were randomly selected for the study. Data were collected through structured observations on the status of compliance with COVID-19 prevention guidelines, and entered using the KoboCollect software, which was preinstalled on mobile devices (smart phones and tablets). Descriptive statistics were generated to measure compliance to the set COVID-19 Ministry of Health prevention guidelines using Stata 14 software.

**Funding:** Funding for this study was provided by the Government of Uganda through the Makerere University Research and Innovations Fund (RIF). The funders had no role in study design, data collection and analysis, decision to publish, or preparation of the manuscript.

**Competing interests:** The authors declare that they have no competing interests.

**Abbreviations:** IEC, Information, education and communication; IPC, Infection prevention and control; SARS-CoV-2, Severe Acute Respiratory Syndrome Coronavirus 2; PPE, Personal protection equipment.

## Results

Only 16.6% (38/229) of the supermarkets complied with the COVID-19 prevention and control guidelines. In line with the specific measures, almost all supermarkets 95.2% (218/229) had hand washing facilities placed at strategic points such as the entrance, and 59.8% (137/229) of the supermarkets surveyed regularly disinfected commonly touched surfaces. Only 40.6% and 30.6% of the supermarkets enforced mandatory hand washing and use of face masks respectively for all customers accessing the premises. Slightly more than half, 52.4% (120/229) of the supermarkets had someone or a team in charge of enforcing compliance to COVID-19 measures and more than half, 55.5% (127/229) of the supermarkets had not provided their staff with job-specific training/mentorship on infection prevention and control for COVID-19. Less than a third, 26.2% (60/229) of the supermarkets had an infrared temperature gun for screening every customer, and only 5.7% (13/229) of the supermarkets captured details of clients accessing the supermarket as a measure to ease follow-up.

## Conclusion

This study revealed low compliance with COVID-19 guidelines, which required mandatory preventive measures such as face masking, regular disinfection, social distancing, and hand hygiene. This study suggests the need for health authorities to strengthen enforcement of these guidelines, and to sensitise the supermarket managers on COVID-19 in order to increase the uptake of the different measures.

## Background

The novel coronavirus disease (COVID-19) remains a significant global health threat, with adverse effects not only on human health, but also trade and industry [1, 2]. This is because trade and industry remain a major source of livelihood for a significant proportion of the global population [3]. Moreover, COVID-19 has impacted the social, physical and psychosocial aspects of communities worldwide [4]. Since the declaration of COVID-19 as a public health threat of international concern on 30th January 2020, the number of cases, and deaths has continued to rise exponentially [5]. As of 23rd July 2021, a total of 192, 284, 207 cases and 4,136,518 deaths were reported globally [5]. In Uganda, the first case of COVID-19 was reported on 21st March 2020. However, prior to the reporting of the first case, the country had started implementing several stringent public health measures to prevent and minimize the spread of COVID-19 [6].

On the 18th of March 2020, the government of Uganda banned all public gatherings for 32 days, and encouraged people in public facilities to: observe the recommended social distance, not to cough, sneeze or spit in public, frequently and appropriately practice hand hygienic (hand washing with soap and water or using alcohol-based hand rub (ABHR)), and regularly clean and disinfect surfaces, such as tables and door handles among others. A number of public facilities such as bars, entertainment centers including discos, cinema halls and sports centers were closed. Additionally, travelers entering the country at the different border points were to be quarantined for 14 days. By the 25th of March 2020, all academic institutions were closed, all border points were closed except for cargo, and a ban on public and non-essential private transport was in place. A 14-day total lockdown with a nationwide curfew from 7 p.m.

to 6.30 a.m. was declared on 30[th] March 2020, and later extended until the 4[th] of June 2020, when a phased easing of the restrictions commenced.

Following the easing of the restrictions, the number of COVID-19 cases increased sharply between June 2020 and June 2021, which forced the government to enforce a second lockdown on the 6[th] of June 2021. As of 23[rd] July 2021, Uganda had registered a total of 91,710 COVID-19 cases and 2,496 deaths. Situational analyses by the Ugandan Ministry of Health indicate that the Kampala Metropolitan area, which includes Kampala City and Mukono Municipality has the highest number of COVID-19 cases in Uganda [6].

Severe Acute Respiratory Syndrome Coronavirus 2 (SARS-CoV-2), the causative agent of COVID-19 is primarily transmitted through respiratory droplets and contact with contaminated surfaces [2, 7]. Surfaces can get contaminated by infected respiratory droplets that are expelled when an infected person coughs, sneezes, laughs or talks [7]. These droplets land on objects and surfaces, which are touched by people who may then touch their eyes, nose or mouth. Once the surfaces are contaminated, the virus may remain viable for hours to days [8]. Currently, there are inconsistent data on the survivability of SARS-CoV-2 on different surface types [9]. However, existing evidence indicates that the virus can survive on plastic, stainless steel and countertops for up to three to four days [10, 11], persists on copper and copper alloys for up to four hours [10, 12], up to two days on glass and up to 24 hours on cardboard [10, 11].

Shopping in supermarkets and other public places regularly visited by many people increases opportunities of transmission of coronavirus. These opportunities for infection spread include frequently touched surfaces such as, refrigerators, door knobs, staircases, elevator buttons, trolleys and basket handles, which may have been contaminated by an infected person. Furthermore, supermarkets have limited spaces that limit social distancing, yet allow frequent human interaction, which heightens the risk of transmission of COVID-19. Owing to this risk, the Ministry of Health (MOH), Uganda developed guidelines to prevent the transmission of COVID-19 in public places including supermarkets. These include; temperature screening of all shoppers and supermarket staff before entry, appropriate wearing of face masks, social distancing, provision of adequate and functional hand washing facilities at strategic points in the supermarket, ensuring hand washing before entry, provision of adequate waste management facilities, regular provision of updated information on COVID-19 to workers, display of posters with information on COVID-19 in conspicuous places and in different languages, and providing a copy of the guidelines on do's and don'ts to every customer who accesses the supermarket premises. Furthermore, the MOH recommends regular cleaning and disinfection of commonly touched surfaces such as, doorknobs, staircase, elevator rails and buttons, counter tops, and communal places such as, bathrooms, toilets, and floor surfaces at least 3–4 times a day under close supervision [13]. Regular cleaning and disinfection of commonly touched and visibly dirty surfaces is critical for the prevention of COVID-19 and other viral respiratory illnesses in public settings [2, 8, 14, 15].

Despite the existence of these guidelines and the public health importance of specific preventive measures, there is limited evidence on the extent of compliance with the COVID-19 prevention guidelines [16]. Some of the previous studies on compliance with COVID-19 prevention guidelines in Uganda have been among rural market vendors [17], among the general population in rural and urban areas [16], and among high risk groups in Kampala [18, 19]. In all these previous studies, generally poor compliance with COVID-19 guidelines was reported. Unfortunately, compliance among supermarkets, which are some of the busiest public places in urban areas has not been studied and implementation of prevention guidelines remains poor. Yet, implementation and enforcement of COVID-19 prevention guidelines can establish norms that protect supermarket workers, shoppers and other clients from infection [20]. This

 

study established the extent of compliance with the COVID-19 prevention guidelines among supermarkets in Kampala Capital City and Mukono Municipality, Uganda.

## Materials and methods

### Study design and area

A cross-sectional observational study utilizing quantitative data collection methods was used to obtain data from supermarkets in Kampala Capital City and Mukono Municipality. Kampala is the capital city of Uganda and has five administrative divisions: Central, Rubaga, Makindye, Kawempe and Nakawa. Kampala's population is estimated at 1.5 million people [21]. Mukono Municipality is 21 kilometres (km) east of Kampala along the Kampala-Jinja Highway. It is bordered by Kalagi to the north, Kira Town to the west, Lake Victoria to the south, and Lugazi to the east. The town occupies approximately 31.4 square km (12.1 sq mi) of land area. Mukono Municipality covers an estimated land area of 210 square km, and is one of the fastest growing municipalities in Uganda. Mukono Municipality is located approximately 20 kilometers east of Kampala. Mukono Municipality consists of 2 Divisions, 9 Wards and 79 Villages/cells. Available data indicate that Mukono has an estimated population of 162,796 inhabitants [22]. Data were collected between 10th– 19th September 2020 during the first wave of COVID-19, and during a period when the country was gradually opening following a 3 months total lockdown, which started in March 2020.

### Study unit, sample size and sampling procedure

The study units were the selected supermarkets in Kampala Capital City and Mukono Municipality. During the sample size calculation, we first obtained the total number of registered supermarkets in the Kampala and Mukono municipality. We then applied the following formula to obtain a representative sample.

$$X = \left[\frac{[(z^2 * p * (1 - p)]}{[(ME^2]}\right.$$

Since there was limited evidence on the level of compliance with COVID-19 guidelines among supermarkets, we assumed a conservative prevalence of compliance of 50%, a 10% non-response rate and a design effect of 1.3, which yielded a minimum sample size of 215 supermarkets.

The selected supermarkets included large, medium and mini supermarkets. During the sampling process, a mini supermarket was defined as a supermarket having 1–2 counter attendants and lacking provisions for sale of fresh meat, a bakery and a store(s) for un-displayed merchandise. A medium sized supermarket was defined as a supermarket having 3–4 counter attendants and lacked provisions for sale of fresh meat, a bakery and a store(s) for un-displayed merchandise, while a large sized supermarket was defined as one having more than 4 counter attendants and had additional provisions for sale of fresh meat, a bakery and a store(s) for un-displayed merchandise. In this study, we included all large supermarkets (19) except 2 which declined to participate. Large supermarkets were purposively selected because they serve the highest volume of clients. A total of 86 medium and 124 mini supermarkets were randomly selected taking into account geographical representativeness and the average number of shoppers served per day (i.e., Kampala Capital City vs. Mukono Municipality). The supermarkets were randomly selected from lists of registered supermarkets (316 Supermarkets in Kampala and 52 supermarkets in Mukono municipality), which were available in the Office of the District Environmental Health Officers in Kampala City and Mukono Municipality.

## Data collection methods and tools

The data was collected through structured observations on the status of compliance with COVID-19 prevention guidelines after review of literature on COVID-19 guidelines and other sources. The English data collection tool (structured observation checklist) is provided in S1 Appendix. Observations were conducted by 12 trained research assistants in the presence of at least a supermarket manager, using a researcher developed digital observation checklist. During the data collection, research assistants observed practices such as social distancing, temperature screening, the use of face masks, availability, functionality and use of hand hygiene facilities, waste management, and presence of information, education and communication materials. In addition, regular screening and frequency of disinfection of commonly touched surfaces were also observed.

## Study variables and measurement

The dependent variable was compliance with guidelines for prevention of COVID-19 in public places. Compliance with infection prevention and control (IPC) measures for supermarkets was defined as "Yes" for those who will have achieved a 75% and above compliance score while the rest were categorized as non-compliant as was used by [23]. Compliance was a composite based on the following indicators: Presence of functional hand hygiene facilities and supplies (hand washing station with water and soap or alcohol based hand rub) at entrances of supermarkets (Yes/No), workers and shoppers observed to be practicing hand-hygiene before entering the supermarket (Yes/No), presence of temperature screening person and equipment at the entrance (Yes/No), temperature measurements observed to be conducted at the entrance (Yes/No), mandatory use of face masks by majority of the workers and shoppers (Yes/No), information, education and communication (IEC) materials on COVID-19 prevention measures displayed at the entrance of the supermarket (Yes/No), and regular disinfection of commonly touched surfaces (Yes/No). For each variable, Yes was scored 1 point while No was scored 0.

## Data management and analysis

Data were entered using the KoboCollect software, which was preinstalled on mobile devices (smart phones and tablets). Data were synchronized onto the online server daily. This allowed for real-time data capture and entry, minimized errors at entry and eased data cleaning [24]. To ensure that the data were secure, only the core study team comprising of the Principal Investigator (RM), Co-investigators (WKK, TS, EB and SK) and the study coordinator had the security key to the KoboCollect server. The data collection tool was evaluated for face and internal validity by a team of experts who are based at the Makerere University School of Public Health. Following field data collection, data were downloaded into an excel file that was later imported into STATA version 14.0 for cleaning and analysis. Data were analyzed using descriptive statistics using means with their corresponding standard deviations, frequencies and proportions. Chi-square tests were done to derive associations between the independent and dependent variable.

## Quality control and assurance

The quantitative data collection tool was designed with data quality checks and skips in order to ensure quality data entry. A team of highly experienced and trained research assistants with a minimum of a Bachelors' degree in either environmental health sciences or social sciences were recruited to undertake the data collection. The research assistants underwent a two days training to equip them with knowledge and skills on community entry, research ethics, and to familiarize with the study protocol. In addition, the data collection tools were pre-tested among 10 supermarkets (2 large, 4 medium and 4 mini supermarkets) in Wakiso district to

detect any possible problems in the flow of the assessment questions, and to establish the actual length of time required to conduct the assessment. At the time of data collection, the research assistants were supervised by the study coordinator, who was in turn accountable to the Principal Investigator. Supervision was aimed at ensuring that the research assistants followed the study protocol when conducting observations and while interviewing the respondents.

### Ethical considerations

This study was approved by the Makerere University School of Public Health (MaKSPH) Higher Degrees, Research and Ethics Committee (HDREC). The study was also reviewed and registered by the Uganda National Council for Science and Technology. Permission to conduct the study was also sought from the relevant authorities (Kampala Capital City Authority and Mukono Municipal Council) and local leaders. Written informed consent was obtained from the supermarket managers before participating in the study. An informed consent document was read to supermarket managers in the appropriate language (either in English or local language) by the research assistants. All information gathered was treated as private. Given that the study was conducted during the COVID-19 pandemic, we developed a risk management plan which was followed by the study team, including the research assistants so as to mitigate any possible risk of infection with COVID-19. Research assistants were required to ensure a social distance while interacting with the respondents, to use face masks and to regularly maintain hand hygiene by hand washing or using an alcohol-based hand rub or sanitizer. Besides, the training of research assistants was done in accordance with the COVID-19 guidelines.

## Results

### Supermarket characteristics

A total of 229 supermarkets were included in this study, including 195 in Kampala Capital City and 34 in Mukono Municipality. More than half, 54.1% (124/229) were mini supermarkets, 37.6% (86/229) were medium sized and 8.3% (19/229) were large. Nearly three quarters, 72.1% (165/229) of the supermarkets had 1–2 counter attendants. Less than a third, 21.0% (48/229) had provisions for sale of fresh meat and about a tenth, 10.9% (25/229) had a bakery (Table 1).

### Compliance with COVID-19 prevention measures

Overall, less than a third, 30.6% (70/229) of the supermarkets enforced mandatory use of face masks for all customers before permitting access to the supermarket premises. Of the supermarkets surveyed, IPC procedures were followed by all customers in only 31.0% (71/229). Slightly more than half, 52.4% (120/229) of the supermarkets had someone or a team in charge of ensuring compliance to COVID-19 measures. Of these, more than half, 62.5% (75/120) had persons that were actively involved in enforcing customer adherence to IPC procedures. Less than half, 44.5% (102/229) of the supermarkets had their staff trained on-job on IPC for COVID-19. Only 5.7% (13/229) of the supermarkets recorded clients accessing the supermarket to enable easy follow-up in case of a suspected case. Nearly two thirds, 62.5% (75/120) of the supermarkets with a team in charge of IPC did not dedicate time to promote hand hygiene at the supermarket. Only 18.8% (43/229) of the supermarkets had provisions for work shifts among the staff. (Table 2).

### Hand hygiene and the use of personal protective equipment (PPE)

Almost all, 95.2% (218/229) of the supermarkets had hand-washing facilities placed at the entrance of the supermarket. Of these, a majority, 97.7% (213/218) had functional facilities. In

**Table 1. Background characteristics of study supermarkets.**

| Description | Attribute | Freq (n = 229) | Percentage (%) |
|---|---|---|---|
| Location | Kampala | 195 | 85.2 |
| | Mukono | 34 | 14.8 |
| Classification of supermarket | Large | 19 | 8.3 |
| | Medium | 86 | 37.6 |
| | Mini | 124 | 54.1 |
| Total number of staff | 0–10 staff | 172 | 75.1 |
| | 11–20 staff | 27 | 11.8 |
| | Over 20 | 30 | 13.1 |
| Number of counter attendants at the supermarket | 1–2 | 165 | 72.1 |
| | 3–4 | 49 | 21.3 |
| | More than 4 | 15 | 6.6 |
| Supermarket has provisions for sale of fresh meat | Yes | 48 | 21.0 |
| | No | 181 | 79.0 |
| Supermarket has a bakery | Yes | 25 | 10.9 |
| | No | 204 | 89.1 |

only 39.3% (90/229) of the supermarkets, all customers washed their hands before entering. Only 16.2% (37/229) of the supermarkets periodically monitored and recorded adherence to

**Table 2. Compliance with COVID-19 prevention measures in supermarkets in Mukono municipality and Kampala Capital City, Uganda.**

| Description | Attribute | Frequency (n = 229) | Percentage (%) |
|---|---|---|---|
| Mandatory hand washing enforced for all customers accessing the supermarket premises | Yes | 93 | 40.6 |
| | No | 136 | 59.4 |
| Mandatory use of face masks enforced for all customers accessing the supermarket premises | Yes | 70 | 30.6 |
| | No | 159 | 69.4 |
| Infection Prevention and control procedures followed by all customers accessing the supermarket premises | Yes | 71 | 31.0 |
| | No | 158 | 69.0 |
| Supermarket has someone or a team in charge of ensuring compliance to COVID-19 preventive measures | Yes | 120 | 52.4 |
| | No | 109 | 47.6 |
| Person/s involved in enforcing customer adherence to infection prevention and control procedures active (n = 120) | Yes | 75 | 62.5 |
| | No | 45 | 37.5 |
| Person/s involved in enforcing customer adherence correctly wearing face masks (n = 120) | Yes | 73 | 60.8 |
| | No | 47 | 39.2 |
| Person/s involved in enforcing customer adherence ensuring social distance among themselves and clients (n = 120) | Yes | 78 | 65.0 |
| | No | 42 | 35.0 |
| Supermarket staff received job-specific training/mentorship on infection prevention and control for COVID-19 | Yes | 102 | 44.5 |
| | No | 127 | 55.5 |
| Record of clients accessing the supermarket is done to enable easy follow-up in case of a suspected case | Yes | 13 | 5.7 |
| | No | 216 | 94.3 |
| Team in charge of IPC have dedicated time to conduct active hand hygiene promotion (e.g., teaching monitoring hand hygiene performance, organizing new activities) (n = 120) | Yes | 45 | 37.5 |
| | No | 75 | 62.5 |
| Supermarket has provisions for work shifts among the staff | Yes | 43 | 18.8 |
| | No | 186 | 81.2 |
| Supermarket staffs given a leave (or offs) | Yes | 102 | 44.5 |
| | No | 127 | 55.5 |

hand hygiene and provided feedback to personnel regarding their performance. Less than a third, 32.3% (74/229) of the supermarkets had posters explaining the steps of hand hygiene. More than half, 64.2% (147/229) of the supermarkets had sufficient and appropriate PPE available and readily accessible to staff. Of these, slightly more than half, 54.4% (80/147) had staff that correctly used PPE. Only 3.5% (8/229) of the supermarkets provided disposable gloves to clients to use during the shopping. Of these, only a quarter, 25% (2/8) reported that all customers use the disposable gloves during their shopping (Table 3).

## Temperature screening and reporting of suspected cases

Less than a third, 26.2% (60/229) of the supermarkets had an infrared thermometer for screening every customer. Of those, 80.0% (48/60) had every customer screened before accessing the

**Table 3. Hand hygiene and use of personal protective equipment in supermarkets in Kampala Capital City and Mukono Municipality, Uganda.**

| Description | Attribute | Frequency (n = 229) | Percentage (%) |
|---|---|---|---|
| **Hand hygiene** | | | |
| Supplies available for ensuring adherence to hand hygiene in the supermarket* | Soap | 201 | 87.8 |
| | Sanitiser | 77 | 33.6 |
| | Alcohol-based hand rub | 67 | 29.3 |
| | Chlorine | 5 | 2.2 |
| | Water | 203 | 88.6 |
| Hand washing facilities placed at the entrance of the supermarket | Yes | 218 | 95.2 |
| | No | 11 | 4.8 |
| Functionality of the Hand washing facility (n = 218) | Yes | 213 | 97.7 |
| | No | 5 | 2.3 |
| All customers wash their hands before accessing the supermarket | Yes | 90 | 39.3 |
| | No | 139 | 60.7 |
| Hand washing facility is well drained (n = 213) | Yes | 201 | 94.4 |
| | No | 12 | 5.6 |
| Supermarket periodically monitors and records adherence to hand hygiene and provides feedback to personnel regarding their performance | Yes | 37 | 16.2 |
| | No | 192 | 83.8 |
| Supermarket has posters explaining the indications for hand hygiene | Yes | 74 | 32.3 |
| | No | 155 | 67.7 |
| Supermarket has posters explaining the correct use of hand rub or hand sanitizer | Yes | 63 | 27.5 |
| | No | 166 | 72.5 |
| Supermarket has posters explaining correct hand washing technique | Yes | 74 | 32.3 |
| | No | 155 | 67.7 |
| **Personal Protective Equipment** | | | |
| Supermarket has sufficient and appropriate PPE available and readily accessible to staff | Yes | 147 | 64.2 |
| | No | 82 | 35.8 |
| PPE correctly used by the supermarket staff (n = 147) | Yes | 80 | 54.4 |
| | No | 67 | 45.6 |
| Supermarket provides disposable gloves to clients to use during the shopping | Yes | 8 | 3.5 |
| | No | 221 | 96.5 |
| All customers use the disposable gloves during their shopping (n = 8) | Yes | 2 | 25.0 |
| | No | 6 | 75.0 |

*multiple responses were captured.

shopping section. Only 35.4% (17/48) of the supermarkets had screeners that were using a flowchart or knew the case definition for COVID-19. Majority, 79.2% (38/48) of the supermarkets had screeners that maintained distance from customers. Only 16.6% (38/229) of the supermarkets performed screening on weekends. Less than a tenth, 5.7% (13/229) of the supermarkets had a designated isolation room/area for suspected cases of COVID-19. Close to half, 49.8% (114/229) of the supermarkets had their staff trained on the use of comprehensive PPE. Less than half, 41.5% (95/229) of the supermarkets had phone/phone credit available for contacting the Ministry of Health COVID-19 surveillance team in case of a suspected case, and 53.7% (123/229) had a list of phone numbers available for staff in the event of a suspect case. More than a third, 60.7% (139/229) of the supermarkets had staff who did not know who to call if a suspected case is identified (Table 4).

## Access to water, sanitation and environmental hygiene

Overall, almost all the supermarkets, 93.0% (213/229) of the supermarkets had access to clean running water. Majority, 90.8% (208/229) of the supermarkets had adequate waste bins. Less than half, 41.9% (96/229) of the supermarkets had COVID-19 posters displayed at the entrance of the supermarket. More than half, 59.8% (137/229) of the supermarkets surveyed regularly disinfected

**Table 4. Temperature screening and reporting of suspected COVID-19 cases in supermarkets in Kampala Capital City and Mukono Municipality, Uganda.**

| Description | Attribute | Frequency (n = 229) | Percentage (%) |
|---|---|---|---|
| Supermarket have an infrared thermometer gun for screening every customer | Yes | 60 | 26.2 |
| | No | 169 | 73.8 |
| Supermarket screens every customer accessing the shopping section using an infrared thermometer (temperature gun) | Yes | 48 | 20.9 |
| | No | 181 | 79.1 |
| Screener/s maintaining distance (at least 2m) from customers **(n = 48)** | Yes | 38 | 79.2 |
| | No | 10 | 20.8 |
| Screener using infrared thermometer gun appropriately (Thermometer held 3–5 cm from temple to get accurate reading) **(n = 48)** | Yes | 47 | 97.9 |
| | No | 1 | 2.1 |
| Screener using flowchart or Screener must be able to apply case definition for COVID19) **(n = 48)** | Yes | 17 | 35.4 |
| | No | 31 | 64.6 |
| Screener wearing appropriate PPE (Using gloves, face shield or goggles, face mask) **(n = 48)** | Yes | 40 | 83.3 |
| | No | 8 | 16.7 |
| Infrared thermometer calibrated? (Look up manufacturer's instructions on internet for calibration instructions) (n = 48) | Yes | 47 | 97.9 |
| | No | 1 | 2.1 |
| Supermarket performs screening on weekends | Yes | 38 | 16.6 |
| | No | 191 | 83.4 |
| Supermarket has a designated isolation room/ area for suspected cases of COVID 19 | Yes | 13 | 5.7 |
| | No | 216 | 94.3 |
| Staff trained on the use of comprehensive PPE | Yes | 114 | 49.8 |
| | No | 115 | 50.2 |
| **Reporting of COVID-19 cases** | | | |
| Phone/phone credit available for contacting the Ministry of Health COVID-19 surveillance team in case of a suspected case | Yes | 95 | 41.5 |
| | No | 134 | 58.5 |
| List of phone numbers available for staff in the event of a suspect case (Must be readily available at the screening station. Staff should have one phone number for notification to avoid needing to call multiple stakeholders) | Yes | 123 | 53.7 |
| | No | 106 | 46.3 |
| Supermarket staff know who to call if a suspect case is identified? | Yes | 139 | 60.7 |
| | No | 90 | 39.3 |

commonly touched surfaces. Almost all, 97.4% (223/229) of the supermarkets had toilet facilities available. Of these, 98.7% (220/223) had toilet structures that were in good status and 85.7% (191/223) had clean sanitary facilities. More than three quarters, 78.0% (174/223) of the supermarkets had toilets with anal cleansing materials. More than three quarters, 88.8% (198/223) of the supermarkets had toilets with hand washing facilities. In more than a third, 38.7% (75/194) of the supermarkets, the hand washing facility at the toilet did not have soap or any disinfectant. About, 88.2% (202/229) of the supermarkets safely stored goods off the floor (Table 5).

## Overall level of compliance among supermarkets

Overall, only 16.6% (38/229) of the supermarkets surveyed were compliant. Majority, 84.8% (161/191) of the non-compliant supermarkets had between 0–10 staff (p = <0.001). More than half, 58.1% (111/191) of the non-compliant supermarkets were mini supermarkets (p = <0.001) (Table 6).

## Discussion

This study sought to assess the extent of compliance with COVID-19 prevention guidelines for public places among supermarkets in Kampala Capital City and Mukono Municipality, Uganda. Overall, this study found that compliance with the Ministry of Health COVID-19 prevention guidelines for public places was sub-optimal, particularly in the mini supermarkets and those with less staff. The proportion of supermarkets where both customers and staff practiced hand hygiene, wore face masks and adhered to social distancing at all times was low, and poses a serious public health threat because strict high-level compliance is required to prevent and control the COVID-19 pandemic. Besides, temperature screening was conducted only in a few of the supermarkets. There was no significant difference in supermarket compliance between those in Kampala and in Mukono.

Less than 20% of the observed supermarkets conducted temperature screening for all shoppers and staff prior to entry. In addition, a majority of the supermarkets reported not conducting temperature screening on weekends. Temperature monitoring is central in the detection of COVID-19 cases since high fever is the most common symptom for the disease [25, 26], and probably the reason it has been recommended in public places including supermarkets. However, only a small proportion of the supermarkets had embraced temperature screening for both shoppers and staff. Failure to conduct temperature screening for all shoppers could be attributed to the lack of infrared thermometers, which characterized most of the supermarkets, and could be attributed to the high cost of IPC supplies, including the infrared thermometers [26]. Besides, temperature monitoring is costly since it requires a dedicated staff [26]. Failure to conduct temperature screening over the weekends could be attributed to the limited enforcement of COVID-19 guidelines by the health authorities over the weekends since public servants rarely work during the weekends. Lack of strict enforcement has widely been reported as a barrier to implementation of disease prevention measures, including those related to COVID-19 [27]. The failure to conduct temperature screening in supermarkets, and particularly over the weekends poses an elevated risk of transmission of COVID-19 since supermarkets usually have higher shopper traffic over the weekends compared to the week days. Our findings therefore, call for reinvigorated public health inspections even on weekends to ensure that the existing COVID-19 measures are implemented in supermarkets.

This study revealed that a majority of the supermarkets had hand washing facilities and adequate waste bins placed at strategic points such as the entrance. The high proportion of supermarkets having hand hygiene facilities placed at strategic points could be attributed to the fact that it was a requirement for them to operate. It is important to note that this study was conducted

**Table 5. Access to water, sanitation and environmental hygiene in supermarkets in Kampala Capital City and Mukono Municipality, Uganda.**

| Description | Attribute | Frequency (n = 229) | Percentage (%) |
|---|---|---|---|
| Supermarket has access to clean running water | Yes | 213 | 93.0 |
| | No | 16 | 7.0 |
| Main source of running water (n = 213) | Piped supply | 206 | 96.7 |
| | Rain water | 7 | 3.3 |
| **Sanitation** | | | |
| Toilet facilities available | Yes | 223 | 97.4 |
| | No | 6 | 2.6 |
| Toilet structure in good status (n = 223) | Yes | 220 | 98.7 |
| | No | 3 | 1.3 |
| Sanitary facilities clean (n = 223) | Yes | 191 | 85.7 |
| | No | 32 | 14.3 |
| Toilet facilities offer adequate privacy (n = 223) | Yes | 219 | 98.2 |
| | No | 4 | 1.8 |
| Toilet facilities have anal cleansing materials (n = 223) | Yes | 174 | 78.0 |
| | No | 49 | 22.0 |
| Toilet facilities have hand washing facilities (n = 223) | Yes | 198 | 88.8 |
| | No | 25 | 11.2 |
| Hand washing facilities functional (n = 198) | Yes | 194 | 98.0 |
| | No | 4 | 2.0 |
| Hand washing facilities within a distance of 5 metres (n = 198) | Yes | 195 | 98.5 |
| | No | 3 | 1.5 |
| Is there evidence of use of functional hand washing facilities (n = 194) | Yes | 191 | 98.5 |
| | No | 3 | 1.5 |
| Hand washing facility has running water (n = 194) | Yes | 193 | 99.5 |
| | No | 1 | 0.5 |
| Hand washing facility has soap or any disinfectant (n = 194) | Yes | 119 | 61.3 |
| | No | 75 | 38.7 |
| **Environmental hygiene** | | | |
| Supermarket has adequate waste bins | Yes | 208 | 90.8 |
| | No | 21 | 9.2 |
| Supermarket environment kept tidy | Yes | 222 | 96.9 |
| | No | 7 | 3.1 |
| Regular cleaning done at the supermarket | Yes | 221 | 96.5 |
| | No | 8 | 3.5 |
| Regular cleaning schedule present at the supermarket | Yes | 187 | 81.7 |
| | No | 42 | 18.3 |
| COVID-19 posters displayed at the entrance of the supermarket | Yes | 96 | 41.9 |
| | No | 133 | 58.1 |
| **Stores** | | | |
| Supermarket has safe storage of goods off the floor | Yes | 202 | 88.2 |
| | No | 27 | 11.8 |
| Supermarket store easily accessible to the staff | Yes | 177 | 77.3 |
| | No | 52 | 22.7 |

during the early phase of the pandemic, in which many businesses were still under lockdown. Therefore, supermarket managers may have abode due to fear of locking down their businesses to curb the spread of COVID-19 [28, 29], and thus losing a source of livelihood or profit.

**Table 6. Level of compliance to COVID-19 guidelines stratified by supermarket characteristics.**

| Variables | Attribute | Level of compliance | | p-value |
|---|---|---|---|---|
| | | Compliant (n = 38) | Non-compliant (n = 191) | |
| District | Kampala | 30 (79.0) | 165 (86.4) | 0.239 |
| | Mukono | 8 (21.1) | 26 (13.6) | |
| Total number of supermarket staff | 0–10 | 10 (26.3) | 161 (84.8) | |
| | 11–20 | 10 (26.3) | 17 (8.9) | <**0.001** |
| | Over 20 staff | 18 (47.4) | 12 (6.3) | |
| Classification of supermarket | Large | 15 (39.5) | 19 (10.0) | |
| | Medium | 10 (26.3) | 61 (31.9) | <**0.001** |
| | Mini | 13 (34.2) | 111 (58.1) | |
| Number of counter attendants | 1–2 | 17 (44.7) | 148 (77.5) | |
| | 3–4 | 15 (39.5) | 34 (17.8) | <**0.001** |
| | More than 4 | 6 (15.8) | 9 (4.7) | |

Although provision of hand hygiene facilities was nearly universal in the surveyed supermarkets, mandatory hand hygiene was not enforced in more than half of the supermarkets. In addition, more than two-thirds did not enforce the mandatory use of face masks for all customers accessing the supermarket premises. The fact that half of the surveyed supermarkets did not enforce mandatory hand hygiene and use of face masks could have been attributed to the low levels of knowledge and a low-risk perception of the supermarket managers. Besides, the fear of losing customers if they are asked to wash hands and to wear face masks could have made some supermarket managers reluctant to institute mandatory hand hygiene and the use of face masks. Failure to ensure mandatory hand hygiene and the use of face masks however, poses a serious health threat not only to the supermarket staff but also to the customers and the community at large [14, 30–32]. This study therefore, highlights the need for the different stakeholders to strictly enforce mandatory hand hygiene and use of face masks in order to reduce the risk of transmission of COVID-19 in supermarkets and protect human health.

Observations conducted in the study supermarkets revealed that a significant percentage (40.2%) of the supermarkets did not conduct regular disinfection of commonly touched surfaces. Although the existing guidelines for the prevention of COVID-19 in public places require the managers of such places to regularly clean and disinfect all communal places such as bathrooms, toilets, floor surfaces; and frequently touched surfaces such as doorknobs/handles with a disinfectant or soap and water [13], it was not the case for supermarkets in Kampala capital City or Mukono Municipality. This could have been attributed to limited resources [27]. Supermarkets, particularly the medium and mini supermarkets could be struggling to break-even implying that they may not have sufficient funds to invest in IPC. Failure to regularly disinfect communal and commonly touched surfaces such as door handles, trolleys and counter tables may elevate the risk of transmission of COVID-19 among the shoppers and the supermarket staff [14]. Therefore, this study calls for strengthened enforcement of implementation of COVID-19 guidelines in supermarkets by health authorities such as the health inspectorate.

More than half of the supermarkets had a dedicated person or team in charge of ensuring compliance to COVID-19 preventive measures. Yet the lack of dedicated persons in charge of IPC in supermarkets could be attributed to the fact that more than half of the staff in the surveyed supermarket had never received job-specific training/mentorship on IPC for COVID-19, and thus could have had limited knowledge. Yet, IPC play a crucial role in the prevention of COVID-19 [33]. Existence of dedicated person or a team in charge of IPC derails sustainability of IPC infrastructure and associated measures [33], which could foster resilience in

future outbreaks/ pandemics. Lack of a team/ someone in charge of IPC in supermarkets could also be a precursor to non-adherence to the existing COVID-19 prevention guidelines such as the use of face masks, social distancing and hand hygiene in supermarkets.

Just half of the supermarkets with a staff member involved in ensuring compliance with COVID-19 preventive measures had them trained in IPC for COVID-19. Failure to orient the compliance staff in IPC may not only elevate their risk to the COVID-19 infection but also that of other staff and the customers. The fact that a significant proportion of the compliance staff was not trained implies that they could have limited knowledge on the transmission dynamics and prevention of COVID-19. It could partly explain for non-masking and failure to observe social distancing among more than a third of the supermarkets' compliance officers at the time of the survey. There is evidence that lack of training for individuals involved in IPC limits their capacity to implement prevention measures [34, 35]. This study therefore, strongly suggests the need to urgently train compliance officers on IPC, particularly as case numbers continue to rise at the time of writing.

Less than a tenth of the supermarkets recorded customers/shoppers contact details, as required for contact tracing in the event of a case or suspected case and less than half of the study supermarkets had phone/phone credit available for calling in case of reporting suspected cases. This could be so because of fear of breach of privacy of the shoppers and the anticipated negative attitude of shoppers towards the practice. Nevertheless, Barnes and Sax [20] point out that the responsible conduct of a business presents an opportunity for public places such as supermarkets to contribute to traditional and innovative disease-control measures, such as contact tracing with the use of mobile applications ("apps") on personal devices. Therefore, failure of the supermarkets to record details of shoppers, and to avail mobile credit to those involved in ensuring compliance derails public health measures such as surveillance, and contact tracing in particular [20].

## Conclusions and recommendations

This study revealed very low compliance with COVID-19 guidelines in selected supermarkets of Kampala Capital City and Mukono Municipality, Uganda, despite requirements to institute mandatory preventive measures such as face masking, regular disinfection, social distancing, and hand hygiene. The proportion of supermarkets where both customers and staff practiced hand hygiene, wore face masks and adhered to social distancing at all times was low, and poses a serious public health risk related to the transmission of COVID-19. Besides, temperature screening was conducted only in a few of the supermarkets. There was no significant difference in supermarket compliance between those in Kampala and in Mukono.

This study therefore, suggests the need for the health authorities to proactively sensitize those involved in the implementation of COVID-19 measures on the importance and mechanisms of implementing IPC. The health authorities also need to strengthen the inspection of public places such as supermarkets to ensure that the recommended measures are implemented at all times, including weekends. The government should put in place penalties for supermarkets that lack the necessary infrastructure and provisions related to hand hygiene, regular cleaning and disinfection, PPE, and social distancing. Where available, the public should make use of the IEC materials and resource persons to learn more about the COVID-19 prevention guidelines for public facilities like supermarkets.

## Supporting information

**S1 Appendix. Structured observation tool for compliance with COVID-19 prevention and control guidelines in supermarkets.**
(DOCX)

**S1 Dataset.**
(XLSX)

## Acknowledgments

We would like to thank the management teams of the respective study supermarkets for allowing each of them to be part of the study. We also remain indebted to the leadership of Mukono Municipality and that of Kampala Capital City Authority for their guidance during the implementation of the survey. We would also like to thank the research assistants who diligently collected the data presented in this manuscript. It was because of your tireless efforts that this study was a success.

## Author Contributions

**Conceptualization:** Richard K. Mugambe, Tonny Ssekamatte, Stevens Kisaka, Esther Buregyeya, Simon Kasasa, Joanne A. McGriff.

**Formal analysis:** Richard K. Mugambe, Stevens Kisaka, Julie Balen, Renuka Kapoor.

**Funding acquisition:** Richard K. Mugambe, Tonny Ssekamatte, Stevens Kisaka, Esther Buregyeya, Simon Kasasa.

**Investigation:** Richard K. Mugambe, Tonny Ssekamatte, Stevens Kisaka, Solomon T. Wafula, John Bosco Isunju, Aisha Nalugya, Patience Oputan, Douglas Kizito Makanga, Michael Mukiibi, Esther Buregyeya, Simon Kasasa, Winnifred K. Kansiime, Joanne A. McGriff.

**Methodology:** Richard K. Mugambe, Tonny Ssekamatte, Stevens Kisaka, Solomon T. Wafula, John Bosco Isunju, Esther Buregyeya, Simon Kasasa, Julie Balen, Renuka Kapoor, Joanne A. McGriff.

**Project administration:** Richard K. Mugambe, Tonny Ssekamatte.

**Supervision:** Tonny Ssekamatte, Stevens Kisaka, John Bosco Isunju, Esther Buregyeya, Simon Kasasa, Winnifred K. Kansiime.

**Writing – original draft:** Richard K. Mugambe, Tonny Ssekamatte, Stevens Kisaka, Solomon T. Wafula, John Bosco Isunju, Aisha Nalugya, Patience Oputan, Douglas Kizito Makanga, Michael Mukiibi, Esther Buregyeya, Simon Kasasa, Winnifred K. Kansiime, Julie Balen, Renuka Kapoor, Joanne A. McGriff.

**Writing – review & editing:** Richard K. Mugambe, Tonny Ssekamatte, Stevens Kisaka, Solomon T. Wafula, John Bosco Isunju, Aisha Nalugya, Patience Oputan, Douglas Kizito Makanga, Michael Mukiibi, Esther Buregyeya, Simon Kasasa, Winnifred K. Kansiime, Julie Balen, Renuka Kapoor, Joanne A. McGriff.

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
