## [Decision Letter · Decision Letter 0]

5 Jul 2021

PONE-D-21-19083

Extent of compliance with COVID-19 prevention and control guidelines among supermarkets in Kampala Capital City and Mukono Municipality, Uganda.

PLOS ONE

Dear Dr. Mugambe,

Thank you for submitting your manuscript to PLOS ONE. After careful consideration, we feel that it has merit but does not fully meet PLOS ONE’s publication criteria as it currently stands. Therefore, we invite you to submit a revised version of the manuscript that addresses the points raised during the review process.

We look forward to receiving your revised manuscript.

Kind regards,

Jianguo Wang, PhD

Academic Editor

PLOS ONE

Journal Requirements:

3. Please include additional information regarding the data collection tool used in the study and ensure that you have provided sufficient details that others could replicate the analyses. For instance, if you developed a tool as part of this study and it is not under a copyright more restrictive than CC-BY, please include a copy, in both the original language and English, as Supporting Information.

Reviewers' comments:

Reviewer's Responses to Questions

**Comments to the Author**

1. Is the manuscript technically sound, and do the data support the conclusions?

Reviewer #1: Yes

2. Has the statistical analysis been performed appropriately and rigorously? 

Reviewer #1: Yes

3. Have the authors made all data underlying the findings in their manuscript fully available?

Reviewer #1: Yes

4. Is the manuscript presented in an intelligible fashion and written in standard English?

Reviewer #1: Yes

5. Review Comments to the Author

Reviewer #1: Dear authors,

Thank you for the paper entitled, “Extent of compliance with COVID-19 prevention and control guidelines among supermarkets in Kampala Capital City and Mukono Municipality, Uganda.” This is a fascinating paper that has tried to address the gap in compliance with COVID-19 guidelines among supermarkets in Uganda. The introduction is well written with a little more need to add some background information. The methods and results are well described. The authors have argued the discussion well. There are few minor comments from my side to be addressed.

1. Page 11, line 8-10: Please update the latest number of cases.

2. Would you mind providing some information on Uganda’s COVID-19 history and how the government has responded to this pandemic in the introduction section?

3. Page 12, line 12-14: “Despite the existence of these guidelines and the public health importance of specific preventive measures, there is limited evidence on the extent of compliance with the COVID-19 prevention 14 guidelines in supermarkets.” Does this limited evidence mean no paper at all? Would you please add a summary of documents complying with COVID-19 guidelines and address your research objective?

4. Page 15: How many research assistants were recruited for the observations?

5. I feel that based on the results and discussion, the conclusion seems a little weak. Could the authors add some solid recommendations and suggestions to the public as well the government to act?

Thank you and good luck.

6. PLOS authors have the option to publish the peer review history of their article (what does this mean?). If published, this will include your full peer review and any attached files.

Reviewer #1: No

---

## [Author Response · Author response to Decision Letter 0]

24 Jul 2021

Editor's comments.

1. Please review your reference list to ensure that it is complete and correct. If you have cited papers that have been retracted, please include the rationale for doing so in the manuscript text, or remove these references and replace them with relevant current references. Any changes to the reference list should be mentioned in the rebuttal letter that accompanies your revised manuscript. If you need to cite a retracted article, indicate the article’s retracted status in the References list and also include a citation and full reference for the retraction notice. - We have reviewed the reference list and all references are correct and complete. No retracted papers have been cited.

2. Please include additional information regarding the data collection tool used in the study and ensure that you have provided sufficient details that others could replicate the analyses. For instance, if you developed a tool as part of this study and it is not under a copyright more restrictive than CC-BY, please include a copy, in both the original language and English, as Supporting Information. 

Thank you. As part of the supporting information, we included a copy of the data collection (English version) tool, that was used. We had the data collection tool in one language (English) since data was collected through observations. The data collection tool was provided in S1 Appendix, see page 7 line 25.

Reviewers' comments

1. Please update the latest number of cases. Thank you for this comment. The latest number of COVID-19 cases was updated as can be seen on page 4 lines 8-9 and 25 -26.

2. Would you mind providing some information on Uganda’s COVID-19 history and how the government has responded to this pandemic in the introduction section? 

Thank you. On page 4, lines 9-26, we provided Uganda’s COVID-19 history and how the government has responded to this pandemic.

3. Despite the existence of these guidelines and the public health importance of specific preventive measures, there is limited evidence on the extent of compliance with the COVID-19 prevention 14 guidelines in supermarkets. Does this limited evidence mean no paper at all? Would you please add a summary of documents complying with COVID-19 guidelines and address your research objective? 

Thank you for this comment. On page 5 line 30 and page 6 lines 1-5, we have added a summary of literature related to the compliance with COVID-19 guidelines in Uganda, and addressed the research gap prior to introducing the objective.

4. How many research assistants were recruited for the observations? 

We included the number of research assistants on page 7 line 26. 

5. I feel that based on the results and discussion, the conclusion seems a little weak. Could the authors add some solid recommendations and suggestions to the public as well the government to act? 

Thank you. On page 20, lines 11-15, the conclusion was strengthened. Besides, on lines 16-24, the recommendations were strengthened.

---

## [Decision Letter · Decision Letter 1]

7 Oct 2021

Extent of compliance with COVID-19 prevention and control guidelines among supermarkets in Kampala Capital City and Mukono Municipality, Uganda.

PONE-D-21-19083R1

Dear Dr. Mugambe,

We’re pleased to inform you that your manuscript has been judged scientifically suitable for publication and will be formally accepted for publication once it meets all outstanding technical requirements.

Kind regards,

Jianguo Wang, PhD

Academic Editor

PLOS ONE

Additional Editor Comments (optional):

Reviewers' comments:

Reviewer's Responses to Questions

**Comments to the Author**

1. If the authors have adequately addressed your comments raised in a previous round of review and you feel that this manuscript is now acceptable for publication, you may indicate that here to bypass the “Comments to the Author” section, enter your conflict of interest statement in the “Confidential to Editor” section, and submit your "Accept" recommendation.

Reviewer #1: All comments have been addressed

2. Is the manuscript technically sound, and do the data support the conclusions?

Reviewer #1: Yes

3. Has the statistical analysis been performed appropriately and rigorously? 

Reviewer #1: Yes

4. Have the authors made all data underlying the findings in their manuscript fully available?

Reviewer #1: Yes

5. Is the manuscript presented in an intelligible fashion and written in standard English?

Reviewer #1: Yes

6. Review Comments to the Author

Reviewer #1: Dear authors,

Thank you for addressing all suggestions. In my opinion, the paper has become more readable and is a good fit for publication in Plos One.

7. PLOS authors have the option to publish the peer review history of their article (what does this mean?). If published, this will include your full peer review and any attached files.

Reviewer #1: No

---

## [Editor Report · Acceptance letter]

21 Oct 2021

PONE-D-21-19083R1 

Extent of compliance with COVID-19 prevention and control guidelines among supermarkets in Kampala Capital City and Mukono Municipality, Uganda 

Dear Dr. Mugambe:

I'm pleased to inform you that your manuscript has been deemed suitable for publication in PLOS ONE. Congratulations! Your manuscript is now with our production department. 

Kind regards, 

on behalf of

Dr. Jianguo Wang 

Academic Editor

PLOS ONE